# Circulation of Dengue Virus Serotype 1 Genotype V and Dengue Virus Serotype 2 Genotype III in Tocantins State, Northern Brazil, 2021–2022

**DOI:** 10.3390/v15112136

**Published:** 2023-10-24

**Authors:** Ueric José Borges de Souza, Ygor da Silva Miranda Macedo, Raíssa Nunes dos Santos, Franciano Dias Pereira Cardoso, Jucimária Dantas Galvão, Evgeni Evgeniev Gabev, Ana Cláudia Franco, Paulo Michel Roehe, Fernando Rosado Spilki, Fabrício Souza Campos

**Affiliations:** 1Bioinformatics and Biotechnology Laboratory, Campus of Gurupi, Federal University of Tocantins, Gurupi 77410-570, Brazil; ygor.apj@gmail.com (Y.d.S.M.M.); engraissanunes@gmail.com (R.N.d.S.); 2Central Public Health Laboratory of the State of Tocantins, Palmas 77054-970, Brazil; francianocardoso@yahoo.com.br (F.D.P.C.); jucydg@mail.uft.edu.br (J.D.G.); 3Department of Physiology and Pathophysiology, Medical University of Sofia, 1431 Sofia, Bulgaria; gabev.evgenig.evgeni@gmail.com; 4Virology Laboratory, Department of Microbiology, Immunology, and Parasitology, Institute of Basic Health Sciences, Federal University of Rio Grande do Sul, Porto Alegre 90050-170, Brazil; anafranco.ufrgs@gmail.com (A.C.F.); proehe@gmail.com (P.M.R.); 5Molecular Microbiology Laboratory, Feevale University, Novo Hamburgo 93525-075, Brazil; fernandors@feevale.br

**Keywords:** DENV, genomic surveillance, molecular clock, arboviruses

## Abstract

In Brazil, the state of Tocantins, located in north-central Brazil, has experienced a significant number of cases of arboviral disease, particularly Dengue virus (DENV). This study aimed to deepen the knowledge on DENV circulation within that state by conducting full genome sequencing of viral genomes recovered from 61 patients between June 2021 and July 2022. There were a total of 8807 and 20,692 cases in 2021 and 2022, respectively, as reported by the state’s Secretary of Health. Nucleotide sequencing confirmed the circulation of DENV serotype 1, genotype V and DENV serotype 2, genotype III in the State. Younger age groups (4 to 43 years old) were mostly affected; however, no significant differences were detected regarding the gender distribution of cases in humans. Phylogenetic analysis revealed that the circulating viruses belong to DENV-1 genotype V American and DENV-2 genotype III Southeast Asian/American. The Bayesian analysis of DENV-1 genotype V genomes sequenced here are closely related to genomes previously sequenced in the state of São Paulo. Regarding the DENV-2 genotype III genomes, these clustered in a distinct, well-supported subclade, along with previously reported isolates from the states of Goiás and São Paulo. The findings reported here suggest that multiple introductions of these genotypes occurred in the Tocantins state. This observation highlights the importance of major population centers in Brazil on virus dispersion, such as those observed in other Latin American and North American countries. In the SNP analysis, DENV-1 displayed 122 distinct missense mutations, while DENV-2 had 44, with significant mutations predominantly occurring in the envelope and NS5 proteins. The analyses performed here highlight the concomitant circulation of distinct DENV-1 and -2 genotypes in some Brazilian states, underscoring the dynamic evolution of DENV and the relevance of surveillance efforts in supporting public health policies.

## 1. Introduction

The Dengue virus (DENV) is an arbovirus transmitted to humans via the bite of infected *Aedes aegypti* mosquitoes [1]. Dengue is an endemic disease in numerous regions worldwide, including tropical and subtropical areas of Asia, Latin America, Africa, and Oceania [2,3]. Taxonomically, the virus belongs to the *Flaviviridae* family, within the genus *Orthoflavivirus* [4]. DENV encompasses four distinct serotypes (DENV-1, DENV-2, DENV-3, and DENV-4) [5]. Within each serotype, the number of genotypes varies; presently, DENV-1 has five genotypes (I–V); DENV-2, six genotypes (I–VI); DENV-3, four genotypes (I–IV); and DENV-4, two genotypes (I–II) [6,7].

The four serotypes of DENV are globally distributed [2,3]. Infections by one serotype grant immunity against that specific serotype but not against the others [8]. Subsequent infections by different serotypes elevate the risk of developing severe forms of the disease [9]. Understanding the potential co-circulation of various virus serotypes can assist in preventing the emergence of more severe diseases [10]. As such, monitoring the circulating DENV genotypes can offer valuable insights into the virus’s evolutionary history and geographic distribution [11].

DENV-1 genotype V predominates in Brazil and probably was introduced in the Americas from Southeast Asia or India [12]. Genotype V has been linked to numerous dengue fever outbreaks in Southeast Asia, including Malaysia, Singapore, and Thailand [12]. Like other DENV-1 genotypes, genotype V can cause a spectrum of symptoms, ranging from mild flu-like illness to severe dengue fever, which can be life-threatening [13]. DENV-2 genotype III, previously named Southern Asian–American, predominates in Brazil [14] and has been associated with various dengue fever outbreaks in South America, especially in Brazil, Venezuela, and Colombia [15]. Similar to other DENV-2 genotypes, genotype III can manifest as a wide range of symptoms, from mild to severe illness, and in certain cases, it can progress to dengue hemorrhagic fever (DHF) or dengue shock syndrome (DSS), which can be life-threatening [16].

Historically, the state of Tocantins in northern-central Brazil has grappled with a high incidence of dengue cases, dating back to the first documented transmission of DENV in the municipality of Araguaína in 1991 [17]. Concurrently, with the advent of the COVID-19 pandemic, SARS-CoV-2 was initially detected in Brazil in late February 2020, reaching the state of Tocantins by March of the same year [18]. Interestingly, alongside the surge of the pandemic virus, there was a noticeable decline in the reporting of suspected dengue cases within the state [19]. However, in 2022, a significant resurgence in dengue cases emerged in Tocantins, with the notification of over 22,000 cases, reinstating the state’s unfortunate status as the leader in dengue cases in the northern region of the country [20].

Just as in other Brazilian states, dengue surveillance in Tocantins falls under the purview of local health authorities, particularly the state and municipal health departments [21]. These agencies are responsible for gathering, consolidating, and analyzing data on reported dengue cases sourced from healthcare providers and laboratories [22]. They also oversee the implementation of measures aimed at controlling the spread of the virus [22]. However, it is important to note that the sequencing of viral genomes typically does not feature within the mandatory protocols established by health authorities [23]. To identify the circulating viral serotypes and genotypes, as well as their dispersion patterns, a more targeted approach involving the sequencing of recovered dengue genomes becomes necessary [11], mainly when the dengue vaccine has been introduced in a country, as is happening at the moment in Brazil [24].

In endemic regions, genomic surveillance significantly contributes to identifying the most probable sources of infection, comprehending geographical distribution, and monitoring the movement of infected vectors [25]. Furthermore, this information plays a pivotal role in establishing early epidemiological markers that guide critical decision-making by competent authorities [23]. Additionally, it aids in determining whether the circulating genotypes are linked to an increased risk of severe dengue (DHF/DSS) within the population [26,27]. Recognizing the importance of this type of analysis, the present study aimed to perform the complete genome sequencing of DENV strains retrieved from Tocantins between June 2021 and July 2022. Additionally, we conducted Bayesian phylogenetic and molecular clock analyses to explore the viral dispersion in Brazil, aiming to provide insight into its spread among Brazilian states and American countries and pinpoint the estimated time of introduction of DENV-1 and DENV-2 into Tocantins.

## 2. Materials and Methods

### 2.1. Ethical Aspects

This project is part of a collaborative initiative between the Central Public Health Laboratory of the Tocantins State Health Department (LACEN-TO/SES-TO) and the Federal University of Tocantins. We used DENV-positive samples generously provided by LACEN/TO, collected between June 2021 and July 2022. These samples, which were used in this investigation, were obtained anonymously from materials exceeding the scope of routine arbovirus identification in Brazilian public health laboratories. All protocols and procedures strictly followed Resolution 466/2012, and the project is registered on Plataforma Brasil under CAAE number 21010719.7.0000.

### 2.2. Sample Collection and RNA Isolation

Sample collections occurred when individuals with suspected dengue sought assistance at a primary healthcare facility. For this study, we collected a total of 128 residual serum clinical samples from individuals exhibiting febrile illness and clinical symptoms indicative of dengue. The collection was carried out by the Secretary of Health of Tocantins between June 2021 and July 2022. It involved patients who voluntarily sought medical care during the epidemic season. These samples were subsequently sent to LACEN-TO. Samples were submitted first to nucleic acid extraction using the QIAmp Viral RNA Mini Kit (Qiagen, Hilden, Germany) and then subjected to real-time reverse transcription PCR to detect ZIKV, CHIKV, and DENV serotypes 1–4 as described previously [28]. Positive samples were selected for sequencing based on two criteria: a cycle threshold value ≤ 25 and the availability of epidemiological metadata, including the date of sample collection, gender, age, and the municipality of residence. All positive samples for DENV-1 (*n* = 57) and DENV-2 (*n* = 4) were subjected to genome sequencing and selected based on their respective RT-qPCR cycle threshold (Ct) values.

### 2.3. Synthesis of cDNA and Multiplex PCR

The extracted RNA was subjected to reverse transcription using Luna Script RT SuperMix (5×) (New England Biolabs, Ipswich, MA, USA). The resulting cDNAs served as templates for whole-genome amplification through multiplex PCR, utilizing the DENV sequencing primer scheme [29], which was divided into two separate pools [30]. Amplicons originating from the two primer pools were consolidated and purified using AMPure XP beads (Beckman Coulter, Brea, CA, USA). The concentrations of the purified PCR products were quantified utilizing a Qubit dsDNA HS Assay Kit with a Qubit 3.0 fluorometer (ThermoFisher Scientific Corporation, Waltham, MA, USA).

### 2.4. Minion Whole Genome Sequencing

The MinION library preparation was conducted employing an SQK-LSK-109 Ligation Sequencing Kit, along with EXP-NBD104 and EXP-NBD114 Native Barcoding Kits (Oxford Nanopore, Oxford, UK). Subsequently, the resultant library was loaded onto Oxford MinION R9.4 flow cells (FLO-MIN106) and subjected to sequencing using a MinION Mk1B device. Raw data collection was facilitated through the ONT MinKNOW software. For high-accuracy base calling of raw FAST5 files and barcode demultiplexing, Guppy (v6.0.1) was employed. Consensus sequences were then generated through de novo assembly utilizing the online tool, Genome Detective [31].

### 2.5. Phylogenetic Analyses

The 57 genome sequences of DENV-1 and the four DENV-2 genomes reported in this study were initially assigned to genotypes using the Dengue Virus Typing Tool [30]. To compare the phylogenetic relationships of the newly generated genome sequences, two datasets were constructed. The dataset for the comparison of DENV-1 genomes comprised 3956 reference sequences of all known five genotypes (I–V) (Appendix A), while the dataset for the comparison of DENV-2 genomes comprised 3211 genomes reference sequences of all known six genotypes (I–VI) (Appendix A). Both datasets were sourced from the National Center for Biotechnology Information [32]. In the selection process, only genomic sequences exceeding 9000 bases in length and accompanied with recorded dates and sampling locations were considered. Sequence alignments were conducted using MAFFT v7.490 with default settings and further reviewed manually using AliView v1.28. Maximum likelihood (ML) phylogenies were constructed using IQ-TREE v2.2.0 [33]. The ML analyses utilized the transition model (TIM2) for nucleotide substitution, incorporating empirical base frequencies (+F), along with the FreeRate model (+R8), which were selected with the ModelFinder software [33]. The analysis comprised 1000 ultrafast bootstrap replicates (−B 1000) and an SH-aLRT branch test (−alrt 1000). Tree visualization was executed using FigTree v1.4.4 [34].

#### Bayesian Evolutionary Analysis

To explore spatial and temporal diffusion patterns, we selected sequences from three distinct DENV-1 clades (Appendix A) and one DENV-2 clade (Appendix A). We conducted multiple sequence alignments using MAFFT with default settings, followed by manual inspection in AliView. ML phylogenies were constructed using IQ-TREE v2.2.0 [33], utilizing the TIM2 + F + R8 substitution model, which was determined as the best-fit model with ModelFinder. The temporal signal assessment was performed with TempEst v1.5.3 [35] through root-to-tip genetic distance regression analysis relative to sampling dates (Appendix A). Once we confirmed sufficient temporal signal in terms of genetic divergence from the root to tip concerning sampling dates, we subjected the aligned sequences to analysis using the BEAST v1.10.4 package [36]. For molecular clock model selection, we employed stringent path sampling (PS) and stepping stone (SS) procedures, running 100 path steps of 1 million interactions each [37].

We employed two molecular clock models: (i) the strict molecular clock model and (ii) the more flexible uncorrelated log-normal relaxed molecular clock model. These were coupled with two non-parametric population growth models: (i) the Bayesian skygrid coalescent model and (ii) the standard Bayesian skyline plot (BSP; 10 groups) [38,39,40]. Both SS and PS estimators consistently favored the Bayesian skyline plot with an uncorrelated log-normal relaxed molecular clock as the most appropriate model for our dataset (Appendix A). We ran three independent MCMC runs, each comprising 50 million states, with sampling every 5000 steps to ensure stationarity and achieve adequate effective sample sizes (ESS) for all statistical parameters. The results from the three independent runs were consolidated using Log Combiner V1.10.4, and MCMC chain convergence was assessed using Tracer v1.7.2 [41]. The maximum clade trees from the MCMC samples were summarized with TreeAnnotator V1.10.4, with 10% burn-in removal, and the resulting MCMC phylogenetic tree was visualized using the ggtree R package [42,43].

We also employed a discrete phylogeographical model [44] to reconstruct the spatial diffusion of the virus across the sampled locations in our dataset. Phylogeographic analyses were performed using an asymmetric model of location transitions. We estimated location diffusion rates through the Bayesian stochastic search variable selection (BSSVS) model, discretized based on location (Brazilian states and countries of the Americas). To ensure reliable results, the MCMC was run for an adequate duration to achieve stationarity and an ESS exceeding 200.

### 2.6. Single Nucleotide Polymorphisms (SNPs) Analysis

To identify missense mutations in the newly sequenced DENV genomes from Tocantins, we initially aligned them using the Minimap2 aligner [45] against the oldest genome within each genotype and serotype. Specifically, DENV-1 genomes were aligned against JQ922544 (DENV-1 genotype V from India, collected in 1963), while DENV-2 genomes were aligned against OK469346 (DENV-2 genotype Southern Asian–American from Jamaica, collected in 1981). The resulting SAM files from the alignments were sorted, converted to BAM format, and indexed using Samtools V1.9 [46]. Subsequently, the BAM file underwent variant detection and genomic VCF file generation through the use of bcftools’ *mpileup* and *bcftools call* functions, both integral components of the Samtools framework [46]. Finally, we applied the bcftools filter to refine the called variations and produce the final VCF file.

## 3. Results

In this study, we recovered near-complete nucleotide sequences of 57 DENV-1 genomes (average coverage: 98%; range: 70.1–99.1%) (Appendix A) plus four DENV-2 genomes (average coverage: 90.7%; range: 89.8–91.3%) (Appendix A). The genomes were recovered from DENV-infected patients in the state of Tocantins, between June 2021 and July 2022. Among the 57 sequenced DENV-1 genomes, 22 (38.6%) were from patients residing in the municipality of Porto Nacional, while the rest were from patients in Palmas (*n* = 13; 22.8%) and other locations (Appendix A and Figure 1). For DENV-2, patients were from Araguaína, Bandeirantes do Tocantins, and Colinas do Tocantins (Appendix A and Figure 1). No significant gender differences were detected among the affected patients, with males comprising 50.8% and females 49.2%. Younger age groups (4 to 43 years old) were the most affected. The mean Ct value at RT-qPCR in the 57 samples from which DENV-1 genomes were recovered was 17.8 (range: 12.1 to 23.1) (Appendix A). For DENV-2 (four genomes), the mean Ct value at RT-qPCR was 21.6 (range: 19.2 to 23.5) (Appendix A).

### 3.1. Phylogenetic Analysis

All 57 DENV-1 genome sequences belonged to genotype V (Figure 2). Additionally, the DENV-1 genomes from Tocantins were observed to group into three distinct clades. Similarly, the four DENV-2 genomes recovered here clustered within genotype III, the so-called Southern Asian–American clade (Figure 3).

#### Bayesian Evolutionary Analysis

The mean evolutionary rate estimated for clade I (or clade Americas, as proposed by Fritsch et al., 2023 [47]) of DENV-1 genotype V was 7.85 × 10^−4^ substitutions per site per year (95% highest posterior density interval: 5.96 × 10^−4^ to 8.63 × 10^−4^). The time of the most recent common ancestor (TMRCA) was estimated to be 20 December 2004 (95% HDP: 12 October 2001, to 15 October 2007). According to the Bayesian phylogeny, the genomes within clade I from the Tocantins state formed a single subclade with strong support (posterior probability = 0.99). This subclade was part of a larger cluster that included sequences from São Paulo, Amazonas, Peru, and Cuba, plus a genome sequence from the USA, isolated from a patient who had traveled to Cuba (OQ445881). The estimated TMRCA for this cluster was 28 October 2010 (95% HDP: 15 September 2007, to 1 November 2018), while for the Tocantins subclade, it was determined to be 12 April 2019 (95% HDP: 18 January 2018, to 22 June 2013) (Figure 4).

Clade II (or clade III according to Fritsch et al., 2023 [47]) of DENV-1 genotype V exhibited a mean evolutionary rate of 9.18 × 10^−4^ substitutions per site per year (95% HDP: 7.55 × 10^−4^ to 1.09 × 10^−3^). The TMRCA for this clade was estimated to be 22 January 2003 (95% HDP: 12 August 2001, to 20 May 2004). The Bayesian phylogeny revealed a close relationship between clade II isolates recovered in the sates of Tocantins and São Paulo (ON426305 and ON632048) (Figure 5). The estimated TMRCA for clade II was 16 May 2017 (95% HDP: 18 September 2014, to 1 November 2018), while the Tocantins subclade’s TMRCA was estimated to be 12 April 2019 (95% HDP: 18 January 2018, to 13 January 2020) (Figure 5). The Bayesian stochastic search variable selection (BSSVS) procedure detected supported rates of diffusion from the São Paulo state to Tocantins with a Bayes Factor of 17.6 and a posterior probability of 0.65.

For DENV-1 genotype V, clade III (clade IV according to Fritsch et al., 2023 [47]), the estimated mean evolutionary rate was 1.06 × 10^−3^ substitutions per site per year (95% HDP: 8.69 × 10^−4^ to 1.16 × 10^−3^). The TMRCA for this clade was estimated to be 20 November 1998 (95% HDP: 29 September 1998 to 31 December 1998). The Bayesian phylogeny indicates that isolates from the Tocantins state form a single, well-supported clade, with a posterior probability of 0.99 (Figure 6). The TMRCA for the Tocantins clade was estimated to be 1 March 2019 (95% HDP: 3 April 2018, to 29 February 2020). Notably, Tocantins strains were closely related to sequences from São Paulo, including one sequence from Florida, USA, associated with travel cases (OQ445880) (Figure 6). The BSSVS procedure identified well-supported rates of diffusion from Goiás state to São Paulo (Bayes Factor: 15.1; Posterior Probability: 0.62) and Tocantins (Bayes Factor: 18.6; Posterior Probability: 0.67).

For the only DENV-2 clade (or lineage BR-4 according to Fritsch et al., 2023 [47] and Santos et al., 2023 [14]) recovered here, the estimated mean evolutionary rate was 1.25 × 10^−3^ substitutions per site per year (95% HDP: 8.83 × 10^−4^ to 1.51 × 10^−3^). The TMRCA was estimated to be 20 March 2016 (95% HDP: 8 February 2015, to 22 February 2017). The Bayesian phylogeny revealed that isolates from Tocantins formed a distinct subclade with strong support, clustering with isolates from Goiás and São Paulo (MT929748 and ON634741, respectively) (Figure 7). The estimated TMRCA for this clade was 26 July 2017 (95% HDP: 5 August 2016, to 22 July 2018), with the Tocantins subclade’s TMRCA estimated to be 16 February 2020 (95% HDP: 14 September 2019, to 19 August 2020). The BSSVS procedure indicated well-supported rates of DENV-2 diffusion from Goiás to Tocantins (Bayes Factor: 75.6; Posterior Probability: 0.92).

### 3.2. SNPs Analysis

The genomic analysis of DENV-1 revealed a total of 122 distinct missense mutations, with 45 of these mutations being shared among the 57 sequenced genomes obtained in this study (Appendix A). The majority of the shared missense mutations were located in the envelope and NS5 proteins. In clade I, which includes two genomes, exclusive point missense mutations were observed in genes coding for *NS1*, *NS2A*, *NS4A*, *2K*, *NS4B*, and *NS5* (Appendix A). Clade II, consisting of eight genomes, exhibited exclusive point missense mutations shared among them in the anchored capsid protein gene, *NS1*, *NS2A*, *NS3*, *NS4A*, and *NS5* (Appendix A). In clade III, encompassing forty-seven genomes, exclusive point missense mutations shared among them were observed in the genes coding for protein pr, envelope protein *E*, *NS1*, *NS2A*, *NS4B*, *and NS5* (Appendix A).

The genomic analysis of DENV-2 revealed a total of 44 distinct missense mutations shared among the four sequences obtained in this study (Appendix A). In addition, some mutations were detected exclusively within specific proteins in particular genomes, such as those highlighted in proteins NS4A and NS5 (Appendix A). Notably, a substantial portion of the identified missense mutations in DENV-2 were concentrated within the NS5 protein, 15 of those shared among all four genomes reported here. It is worth emphasizing that NS5, with approximately 2700 bases, represents the largest protein within DENV genomes. Additionally, significant mutations were observed within the envelope protein, with eight shared missense mutations identified (Appendix A).

## 4. Discussion

Genome sequencing has become pivotal in epidemiological analyses; yet, its availability in developing countries, especially remote regions with limited facilities (though often with frequent outbreaks), remains a challenge [48]. In such a context, samples from suspected DENV cases may endure extended storage in suboptimal conditions and undergo lengthy transportation to reach adequately equipped laboratories, often resulting in sample degradation [49]. To tackle these hurdles, in this study, a portable and relatively less expensive tool, the MinION sequencer, was used throughout. In this study, 57 DENV-1 genomes and 4 DENV-2 genomes were efficiently recovered and fully or near-fully sequenced between June 2021 and July 2022, immediately upon sample collection from patients.

The recent identification of two DENV serotypes in Tocantins aligns with the Ministry of Health’s findings. Out of 25,368 tests conducted in Brazil to detect DENV serotypes up to week 44 of 2022, 84.2% (21,350) were positive for DENV-1, and 15.8% (4018) were positive for DENV-2 [20]. However, in that study, data were primarily gathered to determine prevalent subtypes by focusing on specific regions along the genome. Furthermore, in Tocantins in 2022 [50], a significant rise in dengue cases was detected (18,903 cases in 124 municipalities), underscoring the need for more detailed investigations into the virus’s circulation.

Since the first dengue epidemic in Brazil in 1981 and the initial reports on the detection of DENV in Tocantins in 1991, the virus has been able to perpetuate itself across the state, with reported cases of dengue in humans predominantly concentrated in urban areas [51]. The Brazilian Ministry of Health reported that approximately 70% of documented dengue infections occur in urban areas of municipalities with more than 50,000 inhabitants experiencing economic development [52]. This phenomenon is attributed to commercial exchanges that facilitate the transmission of *Aedes aegypti* and DENV infections [53]. The high population density, availability of mosquito food sources, adaptability of mosquitoes to urban environments, and neglect in controlling mosquito breeding sites are potential factors contributing to this trend [54]. The findings of this study further corroborate that municipalities with populations exceeding 50,000 are the most affected by DENV-1 and DENV-2. Moreover, similar results were observed by Santos et al. (2009) [55] in their investigation of the epidemiological profile of dengue in a neighboring city, Anápolis, in the state of Goiás.

This study reveals that DENV infection can affect people of all age groups, but individuals aged 4 to 43 years old seem more prone to develop symptomatic infections, thus revealing a higher incidence rate for that age group. These findings are consistent with those of Valadares et al. (2012) [56], who analyzed the epidemiological and environmental characteristics of dengue in the two largest cities in the Tocantins over an 11-year period (2000 to 2010). Those authors observed that young adults between 20 and 39 years of age were the most affected [56]. In our study, children and young adults were the age groups in which the infection was most prevalent. Monteiro et al. (2009) [57] found similar results in the city of Teresina, Piauí, Brazil, from 2002 to 2006. They reported that the age group most affected was 15 to 49 years old. Therefore, our study suggests a tendency for dengue infections to occur in younger age groups (4 to 13 and 14 to 23 years old), which could be because young people generally lead more active lifestyles and spend more time outdoors, increasing their exposure to mosquitoes with vector competence for DENV.

### 4.1. Phylogenetic Analysis of DENV-1 and DENV-2

DENV-1 was initially detected in the 1980s in the Northern Region of Brazil, specifically in the state of Roraima. Subsequently, this serotype has spread extensively throughout Brazil [22], including the state of Tocantins, as well as numerous other states across the nation [58]. The sequenced genomes of DENV-1 in this study are classified as genotype V, which includes genomes recovered from the Americas, West Africa, Asia, and the Indian Ocean archipelagos (Comoros, La Reunion, and Seychelles), according to Chen and Vasilakis’ classification [59,60,61,62]. Genotype V has remained the predominant DENV-1 genotype in circulation across the American continent, with genetic makeup reflecting geographical associations [47]. Carvalho et al. (2010) [63] observed that the genetic makeup of DENV-1 strains exhibits geographical associations with the countries from which they were isolated or recovered, given the relatively infrequent movement between countries compared to dissemination within specific countries. In the study conducted by Bruycker-Nogueira et al. (2018) [12], continuous molecular surveillance spanning 30 years of DENV-1 circulation in Brazil affirmed the ongoing prevalence of genotype V in the country.

Additionally, we successfully recovered and fully sequenced four genomes of DENV-2, which were classified as Southeast Asian/American genotype III. These genomes fall into two distinct clades within the phylogenetic tree. One clade comprises strains from Southeast Asia, while the other encompasses strains from Central America, South America, and the Caribbean over the past three decades [59]. The Southeast Asian/American genotype III has been associated with epidemics of DHF/DSS in Latin American countries upon its introduction [64]. Furthermore, this genotype has had a significant epidemiological impact, typically leading to the co-circulation of distinct genotypes of the same serotype in the regions where it is introduced [65]. DENV-2 of the Southeast Asian/American genotype III has been circulating in Brazil since at least the 1990s [66] and has been responsible for several DHF/DSS epidemics in the Americas [67]. This genotype predominates in Brazil at the moment [14,47,68].

#### Bayesian Evolutionary Analysis

The analysis of the phylogenetic tree confirms the distinct presence of DENV-1 genotype V and shows a dynamic viral evolution within the different clades [69,70]. The DENV-1 genomes recovered here exhibit a clear clustering within genotype V, affirming its distinct presence. Additionally, DENV-1 genomes from Tocantins were shown to exhibit a diverse distribution across three separate clades, suggesting a dynamic viral evolution within the region, with several distinct DENV-1 introductions occurring into the state. Similarly, the alignment of DENV-2 genomes with representatives of genotype III Southern Asian–American clade reinforces previous findings [71,72]. Notably, the evolutionary rates within clade I of DENV-1 genotype V were estimated, revealing significant genetic dynamics. In the analysis of genomic sequences from Tocantins within clade I, supported with Bayesian analysis, we observe a broader connection with sequences from various South American regions, including likely travel-related cases. Moving on to clade II of DENV-1 genotype V, the genetic proximity between Tocantins and São Paulo isolates, along with detected diffusion rates, highlights possible dissemination patterns. Furthermore, the strong clade formation amongst Tocantins isolates in clade III underscores their genetic coherence and shared history with São Paulo, and even an international sequence.

A similar pattern emerges for DENV-2, as Bayesian phylogeny shows a distinct Tocantins subclade with strong support, linked to Goiás and São Paulo. Diffusion rates between these states, revealed with the BSSVS procedure, emphasize the complex dynamics of DENV-2 transmission [73]. In summary, these findings shed light on the intricate evolutionary pathways and geographical spread of DENV-1 and DENV-2 in Brazilian states and other Latin American countries, carrying implications that extend to other countries worldwide given travel-related cases. Additionally, these findings highlight the intricate evolutionary pathways and geographical spread of DENV-1 and DENV-2 to Brazilian states and other American countries, including the USA. The diversity and dispersion of these viruses to neighboring states and countries emphasize the ongoing genetic diversity and complexity, underscoring the critical need for continuous genomic surveillance to inform dengue control and prevention strategies.

### 4.2. SNPs Analysis

In the genomic analysis of DENV-1, 122 distinct missense mutations were identified, with 45 being common among 57 sequenced genomes. Notably, the envelope protein, essential for viral entry and progeny virus assembly during cellular infection, displayed seven significant mutations [74]. Equally crucial, the gene encoding the NS5 protein featured seven mutations in 45 genomes. This is of particular importance as NS5 is highly conserved among viral NS proteins and plays a key role in RNA genome capping, methylation, and replication [75]. In light of the imminent large-scale utilization of a Dengue virus vaccine, it is imperative to emphasize the importance of providing comprehensive genomic data regarding viral diversity within specific regions and elucidating its spatial distribution, while identifying potential risk zones. Brazil is currently adopting the Dengue vaccine TAK-003, as highlighted by Rivera et al. (2022) [76]. It is conceivable that specific SNPs within this vaccine may persist and perpetuate through subsequent generations of the virus. Given the immunological pressure exerted by the Dengue vaccine on the vaccinated population, the emergence of new viral mutations is expected [24]. Consequently, the study of these mutations assumes paramount importance in the context of monitoring vaccine efficacy and comprehending the intricate landscape of viral–host co-evolution.

Some limitations need to be considered. Firstly, the accessibility of genome sequencing tools remains a significant challenge, particularly in developing countries and remote areas with limited facilities. Second, the specimens collected from suspected DENV cases are often subjected to unfavorable storage conditions and long transportation distances to better-equipped laboratories, leading to sample degradation that hampers successful sequencing efforts [49]. This study also underscores the potential of portable sequencers, such as the MinION, to address these challenges. This technology can enhance local genomic surveillance efforts, thereby advancing our understanding of the population biology of circulating arboviruses and other emerging pathogens [73]. Additionally, this study focuses on a specific time frame for sample collection (June 2021 to July 2022), which might not fully capture the broader temporal dynamics of DENV spread and evolution. Furthermore, this study’s sample size is relatively small, with 57 DENV-1 genomes and 4 DENV-2 genomes, limiting the generalizability of the findings. Despite these limitations, this study provides valuable insights into the genomic characteristics and spread of DENV-1 and DENV-2 between Brazilian states and other American countries, contributing to the understanding of disease epidemiology and aiding in the formulation of control strategies.

## 5. Conclusions

The current study involved the complete or near-complete genome sequencing of DENV samples from patients from the State of Tocantins, north-central Brazil. This effort provided valuable information on genotype classification and viral dispersion. Compared to usual DENV typing protocols that do not involve full genome sequencing, our approach was shown to be more reliable for genotype identification. Our findings confirm the circulation of DENV serotypes 1 and 2 in Tocantins. Furthermore, phylogenetic analyses of the full or nearly full genomes revealed that the DENV-1 genomes recovered here belong to genotype V American, while the DENV-2 genomes belong to genotype III, Southeast Asian/American clade. It is worth noting that we encountered challenges in reconstructing the transmission path of this viral lineage within the state or the same geographic area due to the limited availability of whole genome sequences for this strain. The scarcity of comprehensive DENV genome sequences across South America significantly hampers our ability to estimate and characterize the molecular epidemiology on a regional scale. A key limitation lies in the absence of sequences from the state before this study. Additionally, we must acknowledge the temporal gaps and uneven distribution of DENV datasets. These factors underscore the importance of implementing genomic surveillance throughout all Brazilian states to combat arboviruses, especially DENV, which is the focus of this study. In conclusion, our analyses reveal the presence of distinct DENV-1 and -2 genotypes and dispersion among Brazilian states, underscoring the dynamic evolution of DENV and the ongoing significance of surveillance efforts in supporting public health policies.

## Figures and Tables

**Figure 1 viruses-15-02136-f001:**
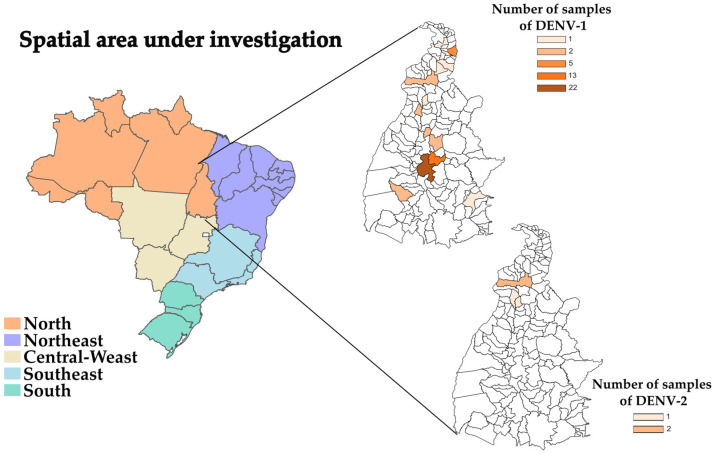
Political map of Brazilian regions with highlights on the Tocantins state, showing locations of the origin of the patients in which the DENV-1 and DENV-2 genomes reported in this study were collected. The gradient colors on the map represent the number of collected samples. For additional information regarding the precise locations of the samples, please consult Appendix A.

**Figure 2 viruses-15-02136-f002:**
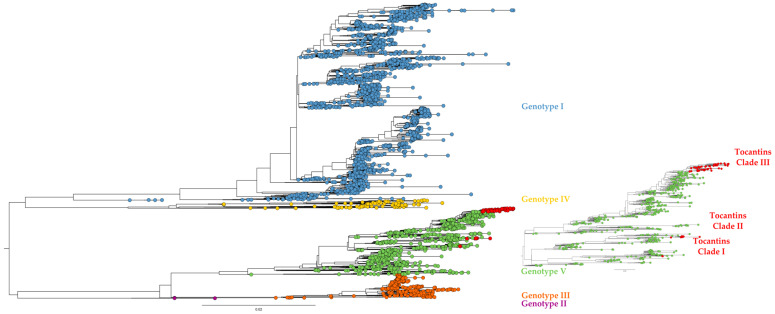
Phylogenetic tree with genomes of all DENV-1 genotypes available at GenBank (accessed on 14 April 2023). Colors indicate the main DENV genotypes. The 57 genomes of DENV-1 reported in the present study are marked with red dots. Genotype I is depicted in blue, genotype II in purple, genotype III in orange, genotype IV in yellow, and genotype V in green.

**Figure 3 viruses-15-02136-f003:**
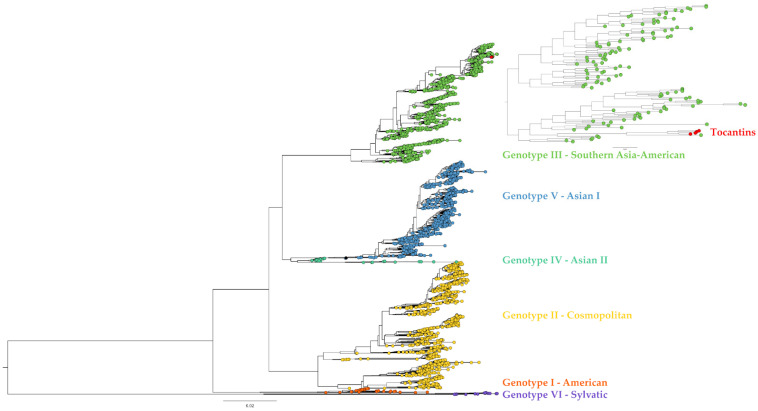
Phylogenetic tree with genomes of all DENV-2 genotypes available at GenBank. Colors indicate the different genotypes. The four genomes of DENV-2 reported in the present study are marked with red dots. Genotype America is depicted in Orange, genotype Cosmopolitan in yellow, genotype Southern Asia–American in green, genotype Asian I in blue, genotype Asian I in light blue and genotype Sylvatic in purple.

**Figure 4 viruses-15-02136-f004:**
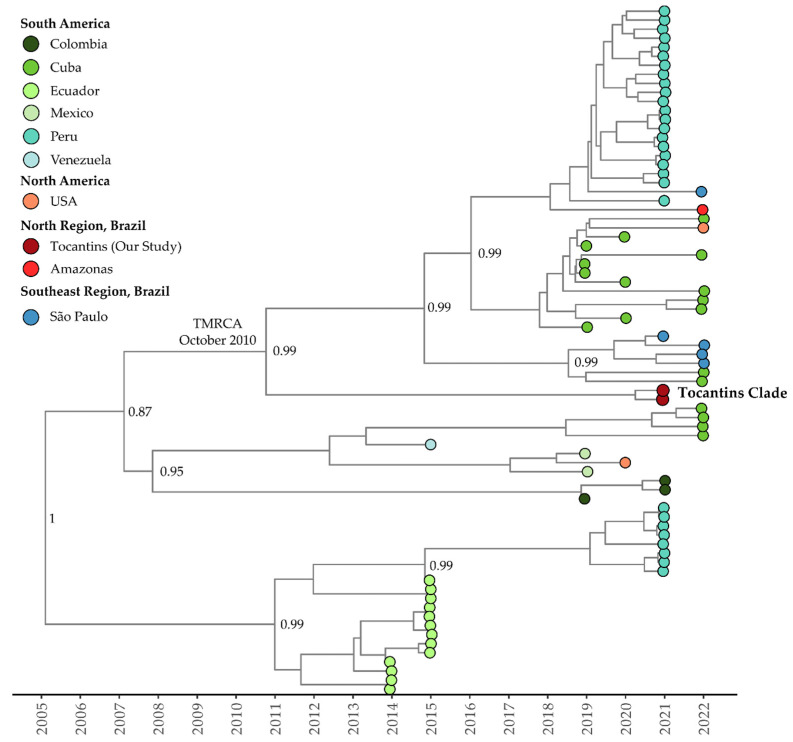
Time-scaled phylogenetic tree of 76 complete and near-complete genomes of DENV-1 genotype 1, clade I recovered in Brazil, South America, and North America. Colors represent different sampling locations according to the legend on the left of the tree. Genomes reported in the presented study are colored in red.

**Figure 5 viruses-15-02136-f005:**
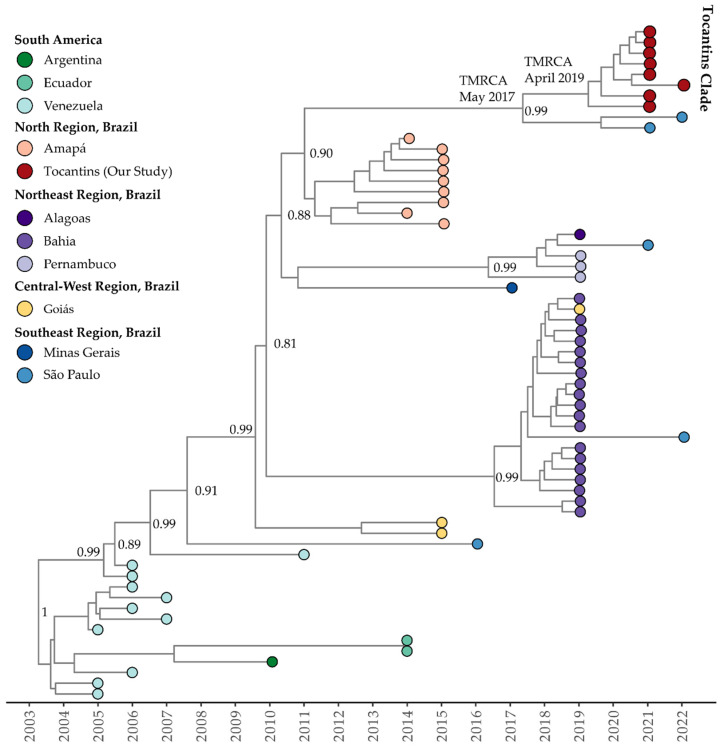
Time-scaled phylogenetic tree of 63 complete or near-complete sequences of DENV-1 genotype V clade II from Brazil and South America. Colors represent different sampling locations according to the legend on the left of the tree. Genomes reported in the presented study are colored in red.

**Figure 6 viruses-15-02136-f006:**
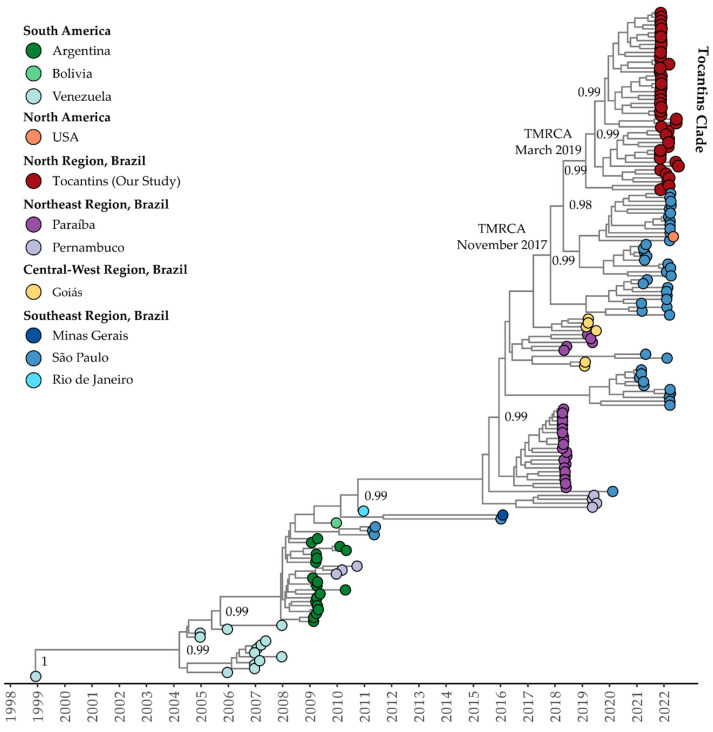
Time-scaled phylogenetic tree of 170 complete or near-complete genomes of DENV-1 genotype V clade III recovered in Brazil, South America, and North America. Colors represent different sampling locations according to the legend on the left of the tree. Genomes reported in the presented study are colored in red.

**Figure 7 viruses-15-02136-f007:**
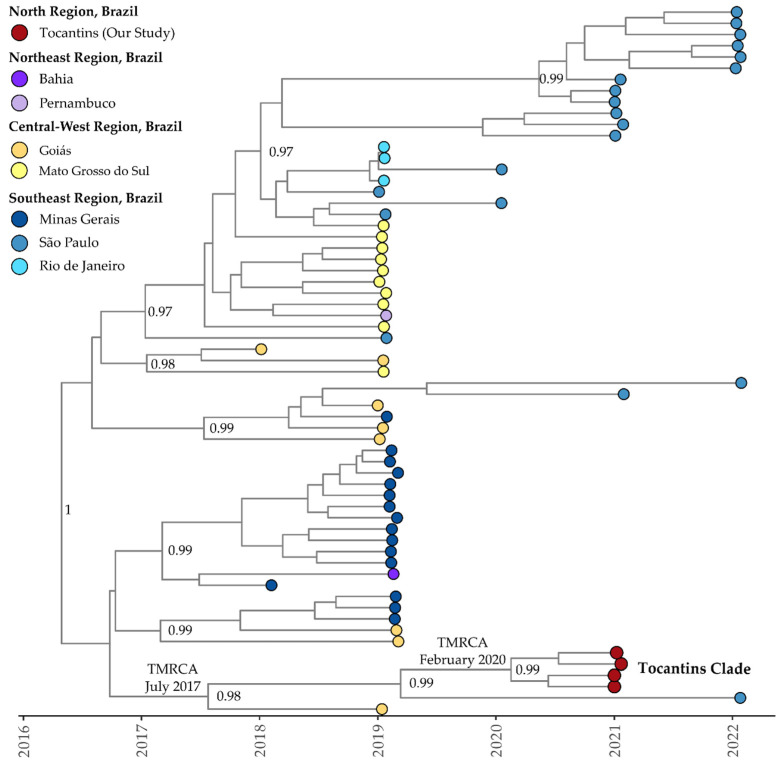
Time-scaled phylogenetic tree of 63 complete and near-complete DENV-2 sequences sampled in Brazil. Colors represent different sampling locations according to the legend on the left of the tree. Genomes reported in the presented study are colored in red.

## Data Availability

The authors declare that all data supporting the findings of this study are available within the paper. The DENV-1 genome sequences have been deposited in GenBank under the accession numbers OR518216 to OR518272. The DENV-2 genome sequences have also been deposited in GenBank with the following accession numbers: OR486032 to OR486035.

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
