# Peer review of "Circulation of Dengue Virus Serotype 1 Genotype V and Dengue Virus Serotype 2 Genotype III in Tocantins State, Northern Brazil, 2021–2022"

_viruses, 2023, doi:10.3390/v15112136_

Round 1

Reviewer 1 Report

Circulation of dengue virus serotype 1 genotype V and dengue virus serotype 2 genotype III in Tocantins State, Northern Brazil, 2011-2022.

Borges de Souza et al.

The authors monitored the circulation of dengue virus (DENV) in Tocantins state by analyzing the viral genome sequences from June 2021 and July 2022. Their findings revealed the presence of both DENV1 and DENV2 among the analyzed DENV-positive patients, with DENV1 being the most predominant serotype. They also reported an equal impact on males and females, with a higher susceptibility observed in the younger age groups. Below are my comments/suggestions:

The Results and Discussion sections require substantial revisions. They should include total number of samples collected (although it was somewhat mentioned in the abstract). Figure 1 appears misleading, as it is unclear what the numbers with color codes represent. In the text, authors state that all 57 DENV1 samples belong to genotype V and are indicated by red dots in Figure 2. However, there seem to be a discrepancy, as red dots are also present in genotype IV in the figure. This inconsistency needs to be addressed and clarified for the readers.

Another problem lies with the SNP analysis. While the authors have identified several mutations in various genes within the DENV genome, it remains unclear what these mutations signify and how they might impact the protein function. To enhance the discussion, it would be beneficial for the authors to provide insights into the specific locations of these mutations within the protein structures and their potential significance. Alternatively, it might be more appropriate to move Tables 1 and 2 to the supplementary section for a more concise and focused main discussion.  

Minor comment: flavivirus has been changed to orthoflavivirus (https://ictv.global/report/chapter/flaviviridae/flaviviridae/orthoflavivirus)

Reviewer 2 Report

Comments

1. In Table S6 and S7 the Ct values were showed, but there was no indication of which RT-qPCR method was applied.

2. Line 25: “spp. “ might be deleted.

3. Line 45: There are too many keywords.

4.Line 444: the reference〔73〕is indicate for the DENV-1, not the DENV-2 circulation in Goias, Brazil.

5.Line 406-407 and 417-419: Authors suggest that at least three independent introduction of DENV-1 genotype V events in Tocantins during the period of 2021-2022, but in reference 12, 14, 47 showed that DENV-1 V clade and DENV-2 III genotype had already persistence in Brazil, the possibility may be the reason from reference 65, that the co-circulation of distinct genotypes of the same serotype in the regions where it is introduced.

Reviewer 3 Report

This study reports the circulation of two serotypes (DENV-1 and DENV-2) in the State of Tocantins; the authors provide the full sequence of 57 strains of DENV-1 and of 4 strains of DENV-2. All the strains were shown to belong to the same genotype (genotype V for DENV-1 and genotype III for DENV-2). For DENV-1 strains, three clades of genotype V were identified, most of them belonging to clade III.

One of the merits of this study is to demonstrate the application of portable sequencers for studying local epidemiology of prevalent viral diseases such as dengue in regions of the world where centers developing high-level genomic technologies are not present. The quality of the genomic analysis is another strength of this study.

This study exhibits some limitations: (i) Despite the fact that the study has the ambition of reporting on-field epidemiology, no clinical data are available, except demographics information; the way the strains were selected for sequencing purpose is not explained; were samples taken from at-random subjects exhibiting low-grade symptoms or from hospitalized patients or both? (ii) There is also no information on the geographical clustering of the cases. Usually, it is interesting to compare the sequences of highly variable regions to monitor the epidemiological link between strains sampled in the same geographic area. This kind of analysis could at least have been done for the numerous strains of clade III of DENV-1 genotype V.

In terms of presentation, the manuscript could be improved. The text is a bit long and could be condensed. The summary is long but not very informative. The first five lines could be removed. The number of sequenced strains is not reported.  The discussion is also too long (3.5 pages); it does not need to repeat the results. In the limitation paragraph, the authors mention the use of a portable sequencer whereas this approach is reported as a strength in the beginning of the discussion. To the reviewer opinion, the lack of clinical and epidemiological data from strains that were sequenced is a true limitation. The conclusion is also too long and appears much more as another summary of the study than an opening to further perspectives. The authors must explain what they plan to do in future dengue outbreaks that could affect their State described as one of the more affected by this arboviral disease. 

Minor remarks:

- line 83: "State" instead of "states"

- footnote of table 1: "within that particular a clade"

- lines 446 and 460: USA are not a part of Latin American countries!

Round 2

Reviewer 1 Report

Circulation of dengue virus serotype 1 genotype V and dengue virus serotype 2 genotype III in Tocantins state, Northern Brazil, 2021-2022.

de Souza et al.

This is an improved version of the previous manuscript. However, I feel that it still requires some modifications. I do not understand what the authors meant by the numbers against different color codes in Figure 1. The Discussion section should be significantly reduced. For example, in section 4.2, the first paragraph (line 290) should be reduced and can be easily merged with the next paragraph (line 510). Line 128 in the Materials and Methods section belongs to the results section. 

Use of abbreviations is inconsistent. For example, Tocantins/TO. Either use ‘Tocantins’ throughout or use ‘TO’ after the first introduction. DENV in the abstract was not defined. 

The version that was provided to me requires significant proofreading. 

The submission of the revised manuscript should follow the same format as the original submission, with double spacing and without the use of  strikethrough.  
